# Association of Health Insurance Status with Outcomes of Sepsis in Adult Patients: A Retrospective Cohort Study

**DOI:** 10.3390/ijerph18115777

**Published:** 2021-05-27

**Authors:** Gaon-Sorae Wang, Kyoung-Min You, You-Hwan Jo, Hui-Jai Lee, Jong-Hwan Shin, Yoon-Sun Jung, Ji-Eun Hwang

**Affiliations:** 1Department of Emergency Medicine, Seoul National University Hospital, 101, Daehak-ro, Jongno-gu, Seoul 03080, Korea; leauuugo@gmail.com (G.-S.W.); loctos00@gmail.com (Y.-S.J.); 2Department of Emergency Medicine, Seoul Metropolitan Government Seoul National University, Boramae Medical Center, 20, Boramae-ro 5-gil, Dongjak-gu, Seoul 07061, Korea; emdrlee@gmail.com (H.-J.L.); skyshiner@naver.com (J.-H.S.); 3Department of Emergency Medicine, Seoul National University Bundang Hospital, 82, Gumi-ro 173 Beon-gil, Bundang-gu, Seongnam 13620, Korea; dyfltk25@naver.com; 4Department of Emergency Medicine, Seoul National University College of Medicine, 103, Daehak-ro, Jongno-gu, Seoul 03080, Korea

**Keywords:** health insurance, sepsis, outcome, mortality

## Abstract

(1) Background: Sepsis is a life-threatening disease, and various demographic and socioeconomic factors affect outcomes in sepsis. However, little is known regarding the potential association between health insurance status and outcomes of sepsis in Korea. We evaluated the association of health insurance and clinical outcomes in patients with sepsis. (2) Methods: Prospective cohort data of adult patients with sepsis and septic shock from March 2016 to December 2018 in three hospitals were retrospectively analyzed. We categorized patients into two groups according to their health insurance status: National Health Insurance (NHI) and Medical Aid (MA). The primary end point was in-hospital mortality. The multivariate logistic regression model and propensity score matching were used. (3) Results: Of a total of 2526 eligible patients, 2329 (92.2%) were covered by NHI, and 197 (7.8%) were covered by MA. The MA group had fewer males, more chronic kidney disease, more multiple sources of infection, and more patients with initial lactate > 2 mmol/L. In-hospital, 28-day, and 90-day mortality were not significantly different between the two groups and in-hospital mortality was not different in the subgroup analysis. Furthermore, health insurance status was not independently associated with in-hospital mortality in multivariate analysis and was not associated with survival outcomes in the propensity score-matched cohort. (4) Conclusions: Our propensity score-matched cohort analysis demonstrated that there was no significant difference in in-hospital mortality by health insurance status in patients with sepsis.

## 1. Introduction

Sepsis is a life-threatening organ dysfunction resulting from infection and is a major healthcare problem. There were approximately 970,000 hospital admissions for sepsis in the United States (US) annually [1]. The number of sepsis cases has been rising over the years, and the costs of sepsis-related hospitalizations are more than $24 billion [1]. Despite improvements in sepsis treatment in line with findings from the Surviving Sepsis Campaign, sepsis accounts for more than 50% of hospital deaths and 30-day mortality remains above 30% among patients with severe sepsis [1,2]. Furthermore, survivors of sepsis may experience prolonged physical and neuropsychological morbidity [3,4], which causes loss of employment or need of caregiver assistance [5].

Various demographic and socioeconomic factors, including age, sex, race, comorbidities, health insurance, residence, and neighborhood, affect disparities in the accessibility of medical treatment and clinical outcomes in sepsis [5,6,7,8,9,10,11,12,13,14,15]. Because sepsis is a lethal disease that requires early and aggressive intense management and medical costs of sepsis treatment are very high [1], socioeconomic factors are important in access to sepsis treatment and outcomes in sepsis. Minejima et al. [11] demonstrated that individuals who lack insurance, who reside in low-income or medically underserved areas, who live far from healthcare, and who lack higher level education face substantial barriers to accessing healthcare and sepsis mortality and hospital readmission is increased in these people.

Health insurance status is one of the important indicators of a patient’s socioeconomic status. It is known to be associated with outcomes of septic patients in the US. In previous studies, patients with Medicare and/or Medicaid or uninsured patients were more likely to have higher mortality than those with private insurance during admission for sepsis [5,6,7,8]. Differences in insurance coverage may be associated with sepsis-related risk factors, including patient age, sex, race, comorbidities, and access to care [5,6,7]. Moreover, an association between health insurance status and outcomes of sepsis itself, independent of known risk factors, would call attention to poor outcomes in sepsis among patients with specific health insurance status and help to explore identifying modifiable factors to improve outcomes of sepsis [5]. However, this has not been fully evaluated in septic patients in countries with a national health insurance system, such as Korea.

Given the increasing numbers of patients with sepsis and expanding costs of sepsis-related hospitalizations, it is important to determine how insurance status affects the outcomes of patients with sepsis. Using multicenter collected hospital-based data and electrical medical records (EMRs), we evaluated the association between health insurance status and clinical outcomes in sepsis. We hypothesized that patients with social health insurance programs would have higher hospital mortality and poorer clinical outcomes in sepsis.

## 2. Materials and Methods

### 2.1. Study Setting and Population

We conducted a retrospective analysis of prospectively collected data of sepsis patients in a multicenter study. Adult patients (age ≥18 years) who were admitted to the emergency department (ED) of three participating hospitals between March 2016 and December 2018 and who met the criteria for sepsis or septic shock were retrospectively analyzed. The three institutions were urban tertiary teaching hospitals, with annual ED visits of approximately 90,000, 70,000, and 50,000 patients. This study was performed in compliance with the Declaration of Helsinki (Fortaleza, Brazil, 2013). Approval for retrospective analysis of the patients was obtained from our institutional review board (BRMH-10-2020-9).

We defined sepsis and septic shock based on the Sepsis-3 definition [16]. Sepsis is defined as life-threatening organ dysfunction caused by a dysregulated host response to infection. Organ dysfunction was identified as an acute change in total Sequential Organ Failure Assessment (SOFA) score ≥2 points. Septic shock was identified with a clinical construct of sepsis with persisting hypotension requiring vasopressors to maintain mean arterial pressure (MAP) ≥65 mmHg and having serum lactate level ≥2 mmol/L (18 mg/dL) despite adequate volume resuscitation. Patients were managed following the international guidelines for the management of sepsis and sepsis shock recommended by the Surviving Sepsis Campaign [2,17].

We excluded patients with a discharge status of “transfer to another hospital” to reduce transfer bias. Patients who had unknown information on their health insurance status or health insurance status coded as “no charge” or “other” and were lost to follow-up were also excluded from the analysis.

The healthcare system in Korea consists of two main insurance programs [18]. National Health Insurance (NHI) covers most residents in Korea, and Medical Aid (MA) supports the lower-income population (NHI 97.2% vs. MA 2.8% in 2019) [19]. Inclusion in NHI or MA is mainly determined by individual socioeconomic status (SES) level. Eligible recipients for MA are persons with recognized income <40% of the standard median income, homeless persons, persons who have national merit, adopted children aged under 18, disaster victims defined under the Disaster Relief Act, and persons and their family members who are governed by the North Korean Refugees Protection and Settlement Support Act [20]. MA is a tax-based program, and the recipients are not charged any monthly premium. On the other hand, NHI recipients pay premiums monthly based on their salary, total property, income, age, and sex.

### 2.2. Data Collection and Outcome Measures

The main exposure was health insurance status. We reviewed the patients’ health insurance status from the EMR systems of the participating hospitals and categorized the patients into 2 groups: the NHI group and the MA group. We extracted demographic data from standardized data collection forms: patient age, sex, comorbidities, initial hemodynamic variables such as blood pressure, heart rate, respiratory rate, and body temperature; source of visit to the ED, primary site of infection, number of septic shocks, SOFA score, and laboratory results such as serum lactate concentration. Do-not-resuscitate (DNR) and against medical advice (AMA) order status and outcome variables such as organ support therapy, intensive care unit (ICU) and hospital length of stay, and mortality were also collected.

The primary outcome was in-hospital mortality. We also evaluated 28-day and 90-day mortality, organ support therapy, length of stay in the ICU, and length of stay in the hospital as secondary outcomes.

### 2.3. Statistical Analysis

Continuous data are expressed as medians and interquartile ranges (IQRs); categorical data are expressed as numbers and percentages. The Mann–Whitney U test, Χ2 test, or Fisher’s exact test were used as appropriate to compare the NHI and MA groups. We performed subgroup analysis to examine the impact of the patients’ health insurance status on in-hospital mortality by septic shock, ICU admission, age, and sex.

We used multivariate logistic regression analysis to evaluate the association between health insurance status and clinical outcomes. Statistically significant variables with *p* < 0.2 from univariable analysis and clinically important variables were included in the final multivariate logistic regression model conducted in a backward stepwise manner. Missing data that exceeded 10% for any variable were not considered in the multivariate logistic regression model. The results of multivariate logistic regression analysis are presented as odds ratios (ORs) with a 95% confidence interval (CI) with a Hosmer–Lemeshow goodness-of-fit test.

To reduce selection bias and potential confounding factors, propensity score analysis was used. The propensity score was calculated using a multivariable logistic regression model, and the following potential confounders were included: patient age, sex, comorbidities, initial mean blood pressure at ED, source of visit to ED, primary site of infection, septic shock, SOFA score, lactate > 2 mmol/L, DNR and AMA order status. Propensity score matching was performed in a one-to-two fashion between the two groups. A standardized mean difference (SMD) ≥ 10% between the two groups was considered to be significant. McNemar’s test for categorical data and the Wilcoxon signed-rank test for continuous data were used to compare the two groups in the propensity score-matched cohort.

*p* < 0.05 was considered statistically significant. All significance levels quoted are two-sided. All data were analyzed using IBM SPSS Statistics version 26.0 software (IBM, Armonk, NY, USA) and R-package 3.3.0 (R Foundation for Statistical Computing, Vienna, Austria).

## 3. Results

### 3.1. Characteristics of Study Patients

A total of 4443 patients with sepsis or septic shock were admitted to the ED during the study period. Patients with follow-up loss (n = 152), those who transferred to another hospital (n = 1667), and those who had unknown information on their health insurance status or health insurance status coded as “no charge” or “other” (n = 98) were excluded from the analysis. Thus, a total of 2526 patients were included in the study (Figure 1).

### 3.2. Comparison between NHI and MA Groups in the Full Cohort

Patient characteristics of the full cohort dichotomized by health insurance status are summarized in Table 1 and Figure 2. Among the eligible population, 2329 patients (92.2%) were covered by NHI, and 197 patients (7.8%) were covered by MA. The MA group had fewer males (NHI vs. MA, 59.7% vs. 50.3%, *p* = 0.009), more chronic kidney disease (7.7% vs. 12.2%, *p* = 0.027), more multiple sources of infection (1.8% vs. 5.1%, *p* = 0.010), and more patients with initial lactate >2 mmol/L (44.1% vs. 59.9%, *p* < 0.001). However, survival outcomes were not different between the NHI and MA groups (Table 2). There were no significant differences in the proportions of patients who received organ support therapy and length of stay in the ICU or hospital between the two groups (Table 2). We also performed subgroup analysis by septic shock, ICU admission, age, and sex (Figure 3). The primary outcome was not different in all subgroups.

### 3.3. Comparison between Survivors and Nonsurvivors and Multivariate Analysis for In-Hospital Mortality

There were 1890 patients with in-hospital survival in the entire cohort. Table 3 shows patient characteristics between survivors and nonsurvivors. Survivors were younger (survivors vs. nonsurvivors, 72.0 years vs. 73.0 years, *p* = 0.012), less likely to be male (56.9% vs. 64.9%, *p* < 0.001), and had higher mean BP (73.0 mmHg vs. 69.0 mmHg, *p* = 0.004). More surviving than nonsurviving patients had community-acquired infection (54.3% vs. 42.9%, *p* < 0.001), fewer surviving than nonsurviving patients had respiratory infection (38.8% vs. 52.5%, *p* < 0.001), fewer surviving patients had shock (18.8% vs. 43.1%, *p* < 0.001), surviving patients had lower SOFA scores (6.0 vs. 8.0, *p* < 0.001), fewer surviving patients had initial lactate > 2 mmol/L (39.6% vs. 62.4%, *p* < 0.001), and fewer surviving patients had documented DNR (5.0% vs. 41.7%, *p* < 0.001). We performed multivariate logistic regression analysis for in-hospital mortality using statistically significant variables with *p* < 0.2 from univariable analysis and clinically important variables, including health insurance status (Table 4). In the multivariate analysis, health insurance status was not independently associated with survival (OR, 0.873; 95% CI, 0.572–1.333) (Table 4). Female sex and genitourinary infection (reference: respiratory infection) were associated with decreased in-hospital mortality, while nosocomial infection, shock status, SOFA score, lactate >2 mmol/L, and DNR status were associated with increased in-hospital mortality.

### 3.4. Propensity Score-Matched Cohort

Using propensity score matching, the patients were matched in a 1:2 ratio, resulting in 384 patients in the NHI group and 192 in the MA group (Table 1). There were no variables that were significantly different between the NHI and MA groups, and standardized mean differences in the propensity-matched cohort were lower than 10% in all variables. Survival and other clinical outcomes were not different between the NHI and MA groups (Table 2).

## 4. Discussion

In the present study, we evaluated an association between health insurance status and clinical outcomes for patients with sepsis. The MA group had fewer males, more chronic kidney disease, more multiple sources of infection, and more patients with initial lactate levels >2 mmol/L. However, clinical outcomes, including in-hospital, 28-day, and 90-day mortality, were not significantly different between the NHI and MA groups. Health insurance status was not independently associated with in-hospital mortality in multivariate analysis and was not associated with survival outcomes in propensity score-matched analysis, adjusting for demographic and clinical factors. Patients with MA for their health insurance were similarly likely to receive organ support therapy compared with those with NHI. There were no significant differences in length of stay in the ICU or hospital between the two groups. Our study suggested that there were no significant disparities in sepsis care and clinical outcomes for patients with sepsis by health insurance status.

The type of health insurance and access to care are known to influence the outcomes in various critical illnesses, including sepsis [5,8,21,22,23]. In prior studies of the relationship of insurance type and outcomes of sepsis in the US, patients with Medicare, Medicaid, or no health insurance had an increased risk of sepsis-associated hospitalization and mortality compared with those with private insurance [5,8]. Kumar et al. [8] reported a 45% increased risk of death during admission for severe sepsis for uninsured patients relative to privately insured patients, adjusting for demographic, clinical, and hospital factors. They suggested that the increased mortality reflects reducing the use of aggressive treatment and intense care for uninsured patients within hospitals as well as delaying or forgoing primary care for potential comorbidities before admission. 

However, our results are quite different from the results of prior studies. Our analysis demonstrated that although patients with MA had more risk factors, such as more chronic kidney disease, more multiple infection sources, and more patients with initial high lactate levels before adjusting, sepsis-associated mortality was not different between patients with NHI and MA. We focused on sepsis treatments performed on both NHI and MA patients as the cause of this result. Substantial intense care and invasive procedures are needed in the treatment of patients with sepsis. Several previous studies have reported the importance of intensive care in sepsis and relationship between intensive care and outcomes in sepsis patients [24,25]. Uninsured patients have been less likely to undergo invasive procedures in previous studies in critical settings [26,27]. However, in our analysis, patients with MA were similarly likely to receive organ support therapy compared with those with NHI.

We also evaluated the impact of study hospitals on the outcomes in patients with sepsis. Haider et al. showed that race and insurance status each independently predicts outcome disparities after trauma [27]. They demonstrated that African American, Hispanic, and uninsured patients had worse outcomes and that mortality was increased in patients admitted to hospitals with a poor payer mix. As uninsured patients were more likely to be admitted to such hospitals, this could contribute to patient mortality. We evaluated each hospital in our analysis with regard to the proportion of patients with NHI and MA and survival outcomes (Appendix A). There were no significant differences in mortality by insurance status in each study hospital.

To our knowledge, this is the first study evaluating an association of health insurance status with mortality for sepsis in countries with a national health insurance system. It is possible that our results, different from prior studies, are likely due to the characteristics of the Korean health insurance system. Most medical care provided during the treatment of sepsis, including organ support therapy and invasive procedures in the ICU, is covered by the national health insurance system in Korea, both in the NHI and MA. In terms of medical cost, after receiving medical treatment and procedures covered by insurance, patients pay a certain portion of the medical costs as copayments. The copayment proportion is usually 5~20% of the total medical cost for patients with NHI and 0~5% for patients with MA when they are admitted to a hospital. The types of medical care and procedures in sepsis treatment covered by both insurance services are similar [20]. Thus, this could contribute to no significant differences in survival outcomes by insurance status in our analysis. In our study, the total medical cost during hospitalization was not different between the NHI and MA groups in a participating hospital (NHI vs. MA, 4683 (2664–8586) vs. 4539 (2248–8427) in US dollars, *p* = 0.628). However, the cost actually charged to patients was much lower in the MA group than in the NHI group (1046 (543–1945) vs. 184 (65–536) in US dollars, *p* < 0.001) (Appendix A).

This study has several limitations. First, this study is a retrospective observational study. Residual confounding factors may exist in the regression model and propensity score-matched cohort. Second, to reduce transfer bias, we excluded patients with a discharge status of “transfer to another hospital”. Thus, in the present study, a large number of patients who transferred to other hospitals were excluded. However, if there was a disparity in the transfer of patients with sepsis based on insurance, our results may include selection bias. Third, we used health insurance status as an index representing patients’ socioeconomic status in the present study. However, it is possible that the patient’s insurance status did not fully reflect the patient’s socioeconomic status. Finally, because this study was a retrospective study analyzing data of three urban teaching hospitals, not including private hospitals, it would be difficult to generalize these results directly to other institutions.

## 5. Conclusions

In our propensity score-matched cohort analysis, there was no significant difference in in-hospital mortality by health insurance status in patients with sepsis. There were no definite disparities in the proportions of patients who received organ support therapy between the NHI and MA groups. The characteristics of the Korean health insurance system could contribute to survival outcomes by insurance status in our analysis. Further confirmation by larger, multicenter studies may provide more precise information on the association of health insurance status with the outcomes of patients with sepsis.

## Figures and Tables

**Figure 1 ijerph-18-05777-f001:**
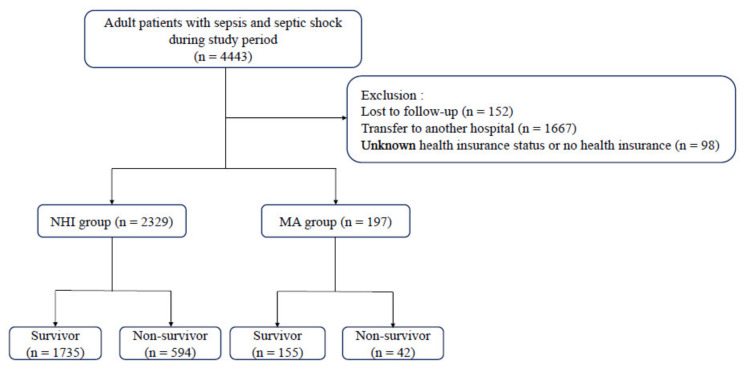
Flow chart of the study population. NHI, National Health Insurance; MA, Medical Aid.

**Figure 2 ijerph-18-05777-f002:**
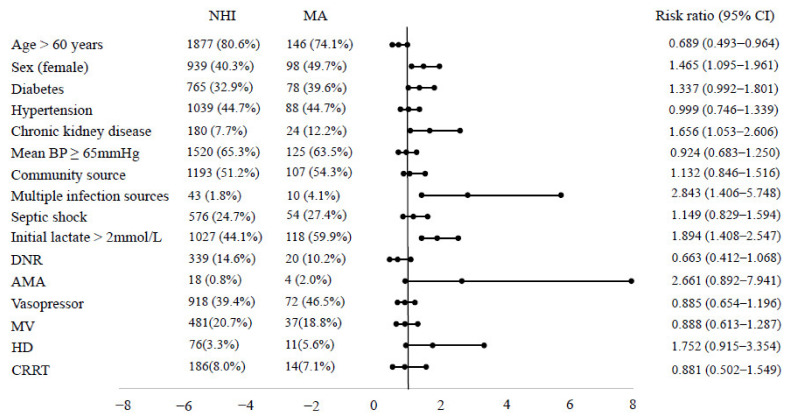
Comparison of NHI versus MA in various variables. The MA group had fewer males (NHI vs. MA, 59.7% vs. 50.3%, *p* = 0.009), more chronic kidney disease (7.7% vs. 12.2%, *p* = 0.027), more multiple sources of infection (1.8% vs. 5.1%, *p* = 0.010), and more patients with initial lactate >2 mmol/L (44.1% vs. 59.9%, *p* < 0.001). NHI, National Health Insurance; MA, Medical Aid; DNR, do-not-resuscitate; AMA, against medical advice; MV, mechanical ventilation; HD, hemodialysis; CRRT, continuous renal replacement therapy; CI, confidence interval.

**Figure 3 ijerph-18-05777-f003:**
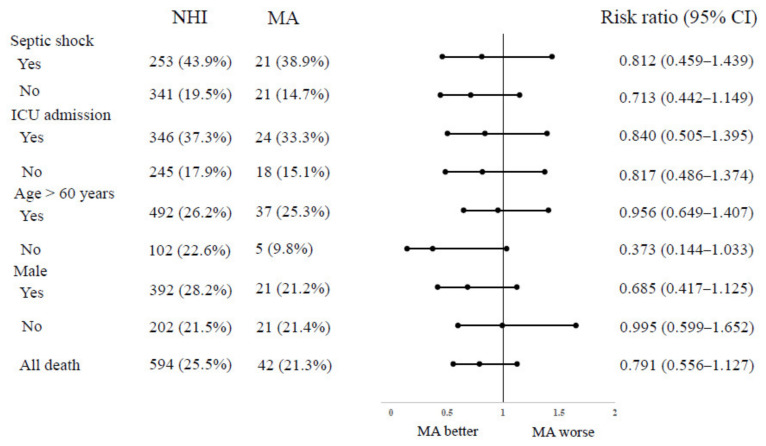
In-hospital mortality by subgroups. We performed subgroup analysis to examine the impact of the patients’ health insurance status on in-hospital mortality by septic shock, ICU admission, age, and sex. In-hospital mortality was not different in all subgroups. NHI, National Health Insurance; MA, Medical Aid.; CI, confidence interval.

**Table 1 ijerph-18-05777-t001:** Patient characteristics of the full study cohort and propensity-matched cohort.

Characteristics	Full Study Cohort				Propensity-Matched Cohort ^a^				
	Total (n = 2526)	National Health Insurance (n = 2329)	Medical Aid (n = 197)	*p*-Value	Total (n = 576)	National Health Insurance (n = 384)	Medical Aid (n = 192)	*p*-Value	SMD
Age (years)	72.0 (63.0–79.0)	73.0 (63.0–79.0)	71.0 (60.0–81.0)	0.293	72.0 (61.3–80.0)	73.0 (63.0–80.0)	71.0 (60.0–81.0)	0.335	0.0625
Sex (male)	1489 (58.9)	1390 (59.7)	99 (50.3)	0.009	288 (50.0)	192 (50.0)	96 (50.0)	1.000	0.0000
Comorbidities									
Diabetes	843 (33.4)	765 (32.9)	78 (39.6)	0.061	247 (42.9)	169 (44.0)	78 (40.6)	0.494	0.0688
Hypertension	1127 (44.7)	1039 (44.7)	88 (44.7)	1.000	268 (46.5)	181 (47.1)	87 (45.3)	0.745	0.0365
Chronic kidney disease	204 (8.1)	180 (7.7)	24 (12.2)	0.027	68 (11.8)	44 (11.5)	24 (12.5)	0.819	0.0314
Initial vital signs									
Mean BP (mmHg)	72.0 (60.0–90.0)	72.0 (60.0–90.0)	74.0 (58.0–96.0)	0.731	74.0 (60.0–93.8)	74.0 (61.0–93.0)	74.5 (58.5–96.0)	0.801	0.0081
Source of visit to ED									0.1358
Community	1300 (51.5)	1193 (51.2)	107 (54.3)	0.416	344 (59.7)	238 (62.0)	106 (55.2)	0.141	
Nosocomial	1121 (44.4)	1042 (44.7)	79 (40.1)	0.231	205 (35.6)	129 (33.6)	76 (39.6)	0.186	
Nursing home	99 (3.9)	89 (3.8)	10 (5.1)	0.442	25 (4.3)	16 (4.2)	9 (4.7)	0.942	
Source of infection									
Respiratory	1067 (42.2)	982 (42.2)	85 (43.1)	0.791	257 (44.6)	172 (44.8)	86 (44.3)	0.976	0.0369
Genitourinary	748 (29.6)	664 (28.5)	64 (32.5)	0.851	189 (32.8)	125 (32.6)	64 (33.3)	0.925	0.0114
Hepatobiliary	235 (9.3)	228 (9.8)	17 (8.6)	0.554	57 (9.9)	40 (10.4)	17 (8.9)	0.657	0.0282
Gastrointestinal	206 (8.2)	186 (8.0)	20 (10.2)	0.294	60 (10.4)	40 (10.4)	20 (10.4)	1.000	0.0089
Others	216 (8.6)	204 (8.8)	12 (6.1)	0.201	41 (7.1)	29 (7.6)	12 (6.3)	0.688	0.0860
Multiple sources	53 (2.1)	43 (1.8)	10 (5.1)	0.010	35 (6.1)	25 (6.5)	10 (5.2)	0.666	0.0584
Septic shock	630 (24.9)	576 (24.7)	54 (27.4)	0.440	171 (29.7)	117 (30.5)	54 (28.1)	0.629	0.0520
SOFA score	6.0 (4.0–9.0)	6.0 (4.0–9.0)	7.0 (4.0–9.0)	0.181	7.0 (4.0–9.0)	7.0 (4.0–9.0)	7.0 (4.0–9.0)	0.837	0.0038
Cardiovascular	1.0 (0.0–4.0)	1.0 (0.0–4.0)	1.0 (0.0–4.0)	0.332	1.0 (0.0–4.0)	1.0 (0.0–4.0)	1.0 (0.0–4.0)	0.163	
Respiratory	2.0 (0.0–2.0)	2.0 (0.0–2.0)	2.0 (0.0–2.0)	0.341	2.0 (0.0–2.0)	2.0 (0.3–2.0)	2.0 (0.0–2.0)	0.811	
Renal	1.0 (0.0–2.0)	1.0 (0.0–2.0)	1.0 (0.0–2.0)	0.064	1.0 (0.0–2.0)	1.0 (0.0–2.0)	1.0 (0.0–2.0)	0.128	
Hepatobiliary	0.0 (0.0–1.0)	0.0 (0.0–1.0)	0.0 (0.0–1.0)	0.018	0.0 (0.0–1.0)	0.0 (0.0–1.0)	0.0 (0.0–1.0)	0.426	
Neurologic	1.0 (0.0–2.0)	0.0 (0.0–2.0)	1.0 (0.0–3.0)	<0.001	1.0 (0.0–2.0)	1.0 (0.0–2.0)	1.0 (0.0–3.0)	0.169	
Coagulation	0.0 (0.0–2.0)	0.0 (0.0–2.0)	0.0 (0.0–1.0)	<0.001	0.0 (0.0–1.0)	0.0 (0.0–1.0)	0.0 (0.0–1.0)	0.473	
Initial lactate > 2 mmol/L	1145 (45.3)	1027 (44.1)	118 (59.9)	<0.001	356 (61.8)	240 (62.5)	116 (60.4)	0.693	0.0425
DNR	359 (14.2)	339 (14.6)	20 (10.2)	0.112	59 (10.2)	39 (10.2)	20 (10.4)	1.000	0.0085
AMA	22 (0.9)	18 (0.8)	4 (2.0)	0.091	11 (1.9)	7 (1.8)	4 (2.1)	1.000	0.0182

Data are presented as median (interquartile range) for continuous variables, and number (%) for categorical variables. ^a^ Propensity score matching was conducted using patient age, sex, comorbidities, initial mean BP at ED, source of visit, source of infection, septic shock, SOFA score, lactate >2 mmol/L, DNR, and AMA order status. SMD, standardized mean difference; BP, blood pressure; ED, emergency department; SOFA, Sequential Organ Failure Assessment; DNR, do-not-resuscitate; AMA, against medical advice.

**Table 2 ijerph-18-05777-t002:** Patient outcomes of the full study cohort and propensity-matched cohort.

	Full Study Cohort				Propensity-Matched Cohort			
	Total (n = 2526)	National Health Insurance (n = 2329)	Medical Aid (n = 197)	*p*-Value	Total (n = 576)	National Health Insurance (n = 384)	Medical Aid (n = 192)	*p*-Value
Primary end point								
In-hospital mortality	636 (25.2)	594 (25.5)	42 (21.3)	0.201	124 (21.5)	82 (21.4)	42 (21.9)	0.971
Secondary end points								
28-day mortality	637 (25.2)	591 (25.4)	46 (23.4)	0.551	135 (23.4)	91 (23.7)	44 (22.9)	0.917
90-day mortality	849 (33.6)	773 (33.2)	76 (38.6)	0.142	198 (34.4)	125 (32.6)	73 (38.0)	0.226
Additional end points								
Organ support therapy								
Vasopressor	990 (39.2)	918 (39.4)	72 (36.5)	0.448	224 (39.0)	152 (39.7)	72 (37.5)	0.677
Mechanical ventilation	518 (20.5)	481 (20.7)	37 (18.8)	0.583	118 (20.5)	81 (21.1)	37 (19.3)	0.688
Conventional hemodialysis	87 (3.4)	76 (3.3)	11 (5.6)	0.100	34 (5.9)	23 (6.0)	11 (5.7)	1.000
Continuous renal replacement therapy	200 (7.9)	186 (8.0)	14 (7.1)	1.000	56 (9.7)	42 (10.9)	14 (7.3)	0.214
ICU length of stay	0.0 (0.0–4.0)	0.0 (0.0–4.0)	0.0 (0.0–4.0)	0.632	0.0 (0.0–4.0)	0.0 (0.0–4.0)	0.0 (0.0–4.0)	0.810
Hospital length of stay	10.0 (5.0–18.0)	10.0 (5.0–18.0)	11.0 (5.5–21.5)	0.101	11.0 (5.0–18.0)	11.0 (5.0–18.0)	11.0 (5.3–21.8)	0.291

Data are presented as median (interquartile range) for continuous variables, and number (%) for categorical variables. ICU, intensive care unit.

**Table 3 ijerph-18-05777-t003:** Patient characteristics stratified as per the primary outcome.

Characteristics	Total (n = 2526)	Survivors (n = 1890)	Nonsurvivors (n = 636)	*p*-Value
Age (years)	72.0 (63.0–79.0)	72.0 (62.0–79.0)	73.0 (65.0–81.0)	0.012
Sex (male)	1489 (58.9)	1076 (56.9)	413 (64.9)	<0.001
Comorbidities				
Diabetes	843 (33.4)	634 (33.6)	209 (33.0)	0.801
Hypertension	1127 (44.7)	858 (45.4)	269 (42.5)	0.200
Chronic kidney disease	204 (8.1)	145 (7.7)	59 (9.3)	0.199
Initial vital signs				
Mean BP (mmHg)	72.0 (60.0–90.0)	73.0 (60.0–91.0)	69.0 (58.0–88.0)	0.004
Source of visit to ED				
Community	1300 (51.5)	1027 (54.3)	273 (42.9)	<0.001
Nosocomial	1121 (44.4)	794 (42.0)	327 (51.4)	<0.001
Nursing home	99 (3.9)	67 (3.5)	32 (5.0)	0.095
Source of infection				
Respiratory	1067 (42.2)	733 (38.8)	334 (52.5)	<0.001
Genitourinary	748 (29.6)	576 (30.5)	172 (27.0)	0.423
Hepatobiliary	235 (9.3)	181 (9.6)	54 (8.5)	0.372
Gastrointestinal	206 (8.2)	139 (7.4)	67 (10.5)	0.011
Others	216 (8.6)	167 (8.8)	49 (7.7)	0.377
Multiple sources	53 (2.1)	38 (2.0)	15 (2.4)	0.596
Septic shock	630 (24.9)	356 (18.8)	274 (43.1)	<0.001
SOFA score	6.0 (4.0–9.0)	6.0 (4.0–8.0)	8.0 (5.0–11.0)	<0.001
Cardiovascular	1.0 (0.0–4.0)	1.0 (0.0–4.0)	3.0 (0.0–4.0)	<0.001
Respiratory	2.0 (0.0–2.0)	1.0 (0.0–2.0)	2.0 (1.0–2.0)	<0.001
Renal	1.0 (0.0–2.0)	0.0 (0.0–1.0)	1.0 (0.0–2.0)	<0.001
Hepatobiliary	0.0 (0.0–1.0)	0.0 (0.0–1.0)	0.0 (0.0–2.0)	<0.001
Neurologic	1.0 (0.0–2.0)	0.0 (0.0–2.0)	1.0 (0.0–3.0)	<0.001
Coagulation	0,0 (0.0–2.0)	0.0 (0.0–1.0)	1.0 (0.0–2.0)	<0.001
Initial lactate >2 mmol/L	1145 (45.3)	748 (39.6)	397 (62.4)	<0.001
DNR	359 (14.2)	94 (5.0)	265 (41.7)	<0.001
AMA	22 (0.9)	17 (0.9)	5 (0.8)	0.790

Data are presented as median (interquartile range) for continuous variables, and number (%) for categorical variables. Groups were compared using Mann–Whitney test or Fisher’s exact test. *p*-values less than 0.05 were considered statistically significant. ED, emergency department; BP, blood pressure; SOFA, Sequential Organ Failure Assessment; DNR, do-not-resuscitate; AMA, against medical advice.

**Table 4 ijerph-18-05777-t004:** Multivariate logistic regression analyses for in-hospital mortality.

Variables	OR	95% CI	*p*-Value
Age (increase by year)	1.003	0.994–1.012	0.569
Sex (female)	0.764	0.603–0.968	0.026
Hypertension	0.840	0.667–1.058	0.138
Mean BP (mmHg)	1.003	0.998–1.009	0.202
Source of visit to ED			
Community	reference		
Nosocomial	1.348	1.082–1.679	<0.001
Source of infection			
Respiratory	reference		
Genitourinary	0.358	0.247–0.518	<0.001
Hepatobiliary	0.744	0.550–1.008	0.056
Gastrointestinal	1.328	0.904–1.951	0.148
Others	0.994	0.664–1.488	0.976
Multiple sources	0.772	0.364–1.635	0.499
Septic shock	1.965	1.506–2.563	<0.001
SOFA score	1.128	1.090–1.166	<0.001
Initial lactate > 2 mmol/L	1.286	0.962–1.719	0.089
DNR	12.001	9.010–15.984	<0.001
Health insurance status	0.873	0.572–1.333	0.530

OR, odds ratio; CI, confidence interval; BP, blood pressure; ED, emergency department; SOFA, Sequential Organ Failure Assessment; DNR, do-not-resuscitate.

## Data Availability

The data that support the findings of this study can be obtained from the corresponding author upon reasonable request.

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
