# Peer review of "Association of Health Insurance Status with Outcomes of Sepsis in Adult Patients: A Retrospective Cohort Study"

_ijerph, 2021, doi:10.3390/ijerph18115777_

Round 1

Reviewer 1 Report

This is a fairly well-conducted and well-written paper. What is lacking is a motivation for the paper. Why would one expect sepsis to be correlated, other things equal, with insurance status? What is it in the Korean system that would predict that patients with private health insurance would be treated differently by doctors/hospitals in the system? In short, what is your theoretical model, what does it predict, and how is the null hypothesis to be tested, statistically?

Author Response

We are attaching our revised manuscript that addresses your comments as follows.

  1. This is a fairly well-conducted and well-written paper.

What is lacking is a motivation for the paper. Why would one expect sepsis to be correlated, other things equal, with insurance status?

Answer> We appreciate your comment. Sepsis is a major healthcare problem, affecting millions of people around the world each year, and its mortality is above 30% among patients with severe sepsis. It requires early and aggressive intensive management represented by sepsis bundle therapy. The medical costs of sepsis-related hospitalization are very high and this may contribute to access to care for septic patients. We added the sentences in Line 49-54 as follows: “Because sepsis is a lethal disease that requires early and aggressive intense management and medical costs of sepsis treatment are very high [1], socioeconomic factors are important access to sepsis treatment and outcomes in sepsis.

Health insurance status is one of the important indicators of patient’s socioeconomic status.”

  1. What is it in the Korean system that would predict that patients with private health insurance would be treated differently by doctors/hospitals in the system? In short, what is your theoretical model, what does it predict, and how is the null hypothesis to be tested, statistically?

Answer> Thank you for your kind comment. In Korea, the healthcare system consists of two main insurance programs, National Health Insurance (NIH) and Medical Aid (MA). In addition to these two basic types of insurance, individuals may optionally purchase a private health insurance. If the patient has an additional private health insurance, the medical costs may be much lower, depending on the coverage of the insurance.

Inclusion in NHI or MA is mainly determined by individual socioeconomic status (SES) level and eligible recipients for MA are usually low SES persons. We hypothesized that patients with MA have more risk factors for sepsis and they are less likely to receive aggressive treatments or procedures which are not covered by their health insurance. So, our initial hypothesis was that patients with MA would have higher hospital mortality and poorer clinical outcomes in sepsis.

To compare in-hospital mortality of septic patients with NHI and those with MA, we performed a retrospective analysis using cohort data of adult patients with sepsis and septic shock and their health insurance status information from electrical medical records (EMRs). Multivariate logistic regression analysis was performed to evaluate the association between health insurance status and in-hospital mortality. Propensity score matching was performed to reduce selection bias and potential confounding factors in retrospective analysis.

Reviewer 2 Report

Wonderful well-written paper with a few minor concerns if the data supports the concerns.

Health disparities are treated differently when it comes to health insurance in the African American, Hispanic, and uninsured communities. What age group in the African American, Hispanic, and uninsured had more sepsis death?  Since this group had a higher mortality rate was it because patients did not want to get treated, physicians did not treat them properly because they were underinsured or using federal aid? The cause of death is very important when it comes to this group because it tells a lot about who is being treated fairly. The Healthcare death rate should be mentioned more in the discussion and methodology section.  

Did this group live in rural areas where hospitals were not familiar or accustomed to treating sepsis?

Lines 261-263-We did not provide an exact explanation for this difference. While unproven by the available data, it is possible that this result is likely because there are no significant differences in in-hospital intensive treatment in either group.  (Why wasn't the difference talked about it's very important and should be mentioned).

Please review APA 7th edition to address to correct the figure. A title is needed. Check references to ensure they are in the correct format. The methodology section is great. https://owl.purdue.edu/owl/research_and_citation/apa_style/apa_formatting_and_style_guide/apa_tables_and_figures.html

Author Response

We are attaching our revised manuscript that addresses your comments as follows.

  1. Wonderful well-written paper with a few minor concerns if the data supports the concerns.

Health disparities are treated differently when it comes to health insurance in the African American, Hispanic, and uninsured communities. What age group in the African American, Hispanic, and uninsured had more sepsis death? Since this group had a higher mortality rate was it because patients did not want to get treated, physicians did not treat them properly because they were underinsured or using federal aid? The cause of death is very important when it comes to this group because it tells a lot about who is being treated fairly. The Healthcare death rate should be mentioned more in the discussion and methodology section. 

Answer> Thank you for your kind comments. Haider et al have shown that outcomes are worse for trauma patients admitted to hospitals with a poor payer mix, such as African American, Hispanic, and uninsured patients. It is unclear whether these patients did not want to get treated, or the physicians did not treat them properly because they were uninsured. However, as a result, these patients are less likely to receive guideline therapy in many critically ill conditions and this leads to poorer outcomes in these patients.

We use this reference to evaluate the hospital effects on sepsis outcome because among our participating institutions, one hospital consists of a relatively higher proportion of MA patients compared with other 2 hospitals. However, in our study, there were no significant differences in mortality by insurance status in each study hospital (We showed sepsis-associated mortalities and proportions of patients by health insurance status in each study hospital in Supplementary data 1).

  1. Did this group live in rural areas where hospitals were not familiar or accustomed to treating sepsis?

Answer> Thank you for your question. Our participating institutions are urban teaching hospitals and patients visit these hospitals from all over the country to treat chronic or critical illnesses. Patients’ residences did not significantly different by insurance status.

  1. Lines 261-263-We did not provide an exact explanation for this difference. While unproven by the available data, it is possible that this result is likely because there are no significant differences in in-hospital intensive treatment in either group. (Why wasn't the difference talked about it's very important and should be mentioned).

Answer> Thank you for your kind comments. We revised the sentences in Line 271-275 as your comments: “We focused on sepsis treatments performed on both NHI and MA patients as the cause of this result. Substantial intense care and invasive procedures are needed in the treatment of patients with sepsis. Several previous studies have reported the importance of intensive care in sepsis and relationship between intensive care and outcomes in sepsis patients [24,25].”

  1. Please review APA 7th edition to address to correct the figure. A title is needed. Check references to ensure they are in the correct format. The methodology section is great. https://owl.purdue.edu/owl/research_and_citation/apa_style/apa_formatting_and_style_guide/apa_tables_and_figures.html

Answer> Thank you for your kind comments. We checked and revised the figures and references as your comments.

Round 2

Reviewer 1 Report

The paper is improved. I still think that, if a more detailed conceptual model could be spelled out clearly, the paper would be improved.

Author Response

Thank you for your helpful comment. We are attaching our revised manuscript that addresses your comments as follows.

The paper is improved. I still think that, if a more detailed conceptual model could be spelled out clearly, the paper would be improved.

Answer> Thank you for your kind comment. We added the sentences according to your comment as follows: “sepsis accounts for more than 50% of hospital deaths” (in Line 43-44) and “Minejima et al [11] demonstrated that individuals who lack insurance, who reside in low-income or medically underserved areas, who live far from healthcare, and who lack higher level education face substantial barriers to accessing healthcare and sepsis mortality and hospital readmission is increased in these people.” (in Line 53-56)
